# T2-High Endotype and Response to Biological Treatments in Patients with *Bronchiectasis*

**DOI:** 10.3390/biomedicines9070772

**Published:** 2021-07-02

**Authors:** Martina Oriano, Andrea Gramegna, Francesco Amati, Alice D’Adda, Michele Gaffuri, Marco Contoli, Francesco Bindo, Edoardo Simonetta, Carlotta Di Francesco, Martina Santambrogio, Giovanni Sotgiu, Francesco Blasi, Stefano Aliberti

**Affiliations:** 1IRCCS Ca’ Granda Ospedale Maggiore Policlinico, Respiratory Unit and Cystic Fibrosis Adult Center, 20122 Milan, Italy; martina.oriano@unimi.it (M.O.); andrea.gramegna@unimi.it (A.G.); alice.dadda@policlinico.mi.it (A.D.); francesco.bindo@unimi.it (F.B.); edoardo.simonetta@unimi.it (E.S.); carlotta.difrancesco@unimi.it (C.D.F.); msantambrogio@gmail.com (M.S.); francesco.blasi@unimi.it (F.B.); 2Department of Pathophysiology and Transplantation, Università degli Studi di Milano, 20122 Milan, Italy; 3Department of Biomedical Sciences, Humanitas University, Pieve Emanuele, 20090 Milan, Italy; francesco.aamati@gmail.com; 4IRCCS Humanitas Research Hospital, Respiratory Unit, Via Manzoni 56, 20089 Rozzano, Milan, Italy; 5Fondazione IRCCS Ca’ Granda Ospedale Maggiore Policlinico, Department of Otolaryngology and Head and Neck Surgery, 20122 Milan, Italy; michele.gaffuri@policlinico.mi.it; 6Department of Clinical Sciences and Community Health, University of Milan, 20122 Milan, Italy; 7Respiratory Unit, Department of Translational Medicine, University of Ferrara, 44121 Ferrara, Italy; ctm@unife.it; 8Clinical Epidemiology and Medical Statistics Unit, Department of Medical, Surgical and Experimental Sciences, University of Sassari, 07100 Sassari, Italy; gsotgiu@uniss.it

**Keywords:** Bronchiectasis, T2-high, type 2 inflammation, eosinophilia

## Abstract

Although bronchiectasis pathophysiology has been historically understood around the presence of airway neutrophilic inflammation, recent experiences are consistent with the identification of a type 2 inflammation (T2) high endotype in bronchiectasis. In order to evaluate prevalence and clinical characteristics of bronchiectasis patients with a T2-high endotype and explore their response to biologicals, two studies were carried out. In a cross-sectional study, bronchiectasis adults without asthma underwent clinical, radiological, and microbiological assessment, along with blood eosinophils and oral fractional exhaled nitric oxide (FeNO) evaluation, during stable state. Prevalence and characteristics of patients with a T2- high endotype (defined by the presence of either eosinophils blood count ≥300 cells·µL^−1^ or oral FeNO ≥ 25 dpp) were reported. A case series of severe asthmatic patients with concomitant bronchiectasis treated with either mepolizumab or benralizumab was evaluated, and patients’ clinical data pre- and post-treatment were analyzed up to 2 years of follow up. Among bronchiectasis patients without asthma enrolled in the cross-sectional study, a T2-high endotype was present in 31% of them. These patients exhibited a more severe disease, high dyspnea severity, low respiratory function, and high impact on quality of life. Among the five patients with severe eosinophilic asthma and concomitant bronchiectasis included in the series, treatment with either mepolizumab or benralizumab significantly reduced the exacerbation rate with an effect that persists for up to 2 years of follow up. If validated across different settings, our data suggest the need to design randomized controlled trials on biological treatments targeting the T2-high endotype in bronchiectasis patients.

## 1. Introduction

Bronchiectasis is a chronic respiratory disease caused by genetic or acquired conditions, characterized by an abnormal and permanent dilation of the bronchi [1]. Bronchiectasis is heterogeneous in terms of radiological and inflammatory patterns, microbiology, patients’ characteristics, and clinical outcomes [1]. It still represents an unmet medical need, particularly in term of response to treatments.

In the last decade, significant effort has been placed into the identification of potential pharmacological targets able to improve the responsiveness to pharmacological interventions in patients with severe chronic respiratory diseases [2]. Notably, biological therapies have been implemented in clinical practice to target molecules involved in the type 2 (T2) inflammatory response in severe asthma patients characterized by a persistent and predominant T2-oriented inflammation [2,3]. Interleukin (IL)-4, IL-5, and IL-13 mediate the T2 inflammatory cascade but are not routinely evaluated [4]. Eosinophil counts in the peripheral blood and oral fractional exhaled nitric oxide (FeNO) are surrogates of the airway eosinophilic inflammation and of the T2-high endotype [5]. Monoclonal antibodies (e.g., mepolizumab and benralizumab, which block IL-5 and IL-5 receptor -IL5ra-, respectively) can reduce the exacerbation rate and improve the quality of life and are able to spare systemic corticosteroid in eosinophilic severe asthmatic patients with bronchiectasis [6].

Up to one-third of bronchiectasis patients show a predominant eosinophilic rather than neutrophilic airway inflammation [7]. High levels of FeNO in bronchiectasis are associated with blood eosinophilia, IL13-driven inflammation, and reduced lung function [7,8]. Furthermore, the level of blood eosinophils seems to identify the subgroup of bronchiectasis patients who can be effectively treated by inhaled corticosteroids (ICS) [5,9,10]. Finally, anti-IL5 or anti-IL5ra drugs result in successful outcomes in severe eosinophilic asthmatic patients with concomitant bronchiectasis [6,11]. In view of these preliminary findings, a better characterization of the T2-high endotype in bronchiectasis patients in terms of disease severity, clinical features, and response to biologicals is warranted. 

The aim of the present research project was to evaluate prevalence, disease severity, and clinical characteristics of bronchiectasis patients showing a T2-high endotype and explore their clinical response to biological drugs. Two different studies were conducted: An observational, cross-sectional study focused on the prevalence and the description of clinical characteristics of bronchiectasis patients with T2-high phenotype, and a case-series of bronchiectasis patients with severe eosinophilic asthma treated for at least one year with either benralizumab or mepolizumab.

## 2. Materials and Methods

### 2.1. Observational, Cross-Sectional Study

An observational, cross-sectional study was conducted at an Italian tertiary care center from September 2016 to September 2019. Consecutive adults (≥18 years) with a clinical (daily sputum production) and radiological (≥1 lobe involvement on high-resolution CT scan) significant bronchiectasis were enrolled during their clinical stability (at least one month apart from the last exacerbation and antibiotic course). Patients with either cystic fibrosis (CF) or traction bronchiectasis due to pulmonary fibrosis were excluded. The study was approved by the local ethical committee, and all subjects provided a written informed consent.

#### 2.1.1. Data Collection

Demographics, functional, radiological, microbiological, and quality of life data were collected during the enrolment. All patients underwent a comprehensive analysis of their pulmonary function including plethysmography with bronchodilator reversibility testing (in case a diagnosis of obstructive ventilatory pattern was made). All patients with a clinical history suggestive for asthma underwent methacholine challenge test according to the European Respiratory Society (ERS) guidelines; furthermore they underwent screening for allergic bronchopulmonary aspergillosis as recommended by the ERS guidelines [12].

All patients underwent both blood eosinophil counts and oral FeNO evaluation during stable state at baseline. FeNO was determined using the Analyzer CLD 88 sp FeNO analyzer (Eco Medics AG, Durnten, Switzerland) according to the ERS recommendations [13]: Patients were asked to inhale the maximum amount of air and then to exhale the air into the valve. The flow rate (50 mL/s) was kept constant, and data were recorded after 90 s.

Asthma was diagnosed according to the latest international guidelines [14]. Severity of bronchiectasis was evaluated according to the Bronchiectasis Severity Index (BSI) [15]. Radiological severity was assessed through the modified Reiff score [16]. Bacteriological examination was performed on spontaneous sputum samples [17]. Chronic infection was defined by the isolation of potentially pathogenic bacteria in sputum culture on two or more times, at least 3 months apart over a 1-year period [18].

#### 2.1.2. Study Groups and Definitions

T2-high endotype was defined by the evidence of either blood eosinophil counts ≥300 cells·µL^−1^ or oral FeNO ≥ 25 dpp [19]. Based on this definition, patients were divided into two study groups: T2-high with a blood eosinophil counts ≥300 cells·µL^−1^ and/or oral FeNO ≥ 25 dpp; non-T2-high group with both blood eosinophil counts <300 cells·µL^−1^ and oral FeNO < 25 dpp. Analysis was conducted on the entire study cohort and on bronchiectasis patients without a concomitant diagnosis of asthma.

#### 2.1.3. Statistical Analysis

Statistical analysis was conducted using R (R Core Team, 2020) version 4.0.0. Qualitative variables were summarized with absolute and relative (percentage) frequencies. Quantitative variables were summarized with medians (interquartile ranges, IQR) in case of non-parametric distribution. Qualitative variables were compared with chi-squared test. Quantitative variables were compared with the Mann–Whitney test in case of non-parametric distribution. A two-tailed *p*-value less than 0.05 was considered statistically significant.

### 2.2. Case-Series

Bronchiectasis patients with severe asthma who were exposed for at least 12 months to anti-IL5 or anti-IL5ra monoclonal antibodies (either mepolizumab or benralizumab) from September 2019 to April 2020 were described. Patients underwent clinical evaluation before treatment and after 6, 12, and 24 months of treatment. Demographics, disease severity (evaluated through the BSI), number of exacerbations and of hospitalizations during the year prior to treatment initiation and during follow up, sputum volume, blood eosinophil counts, number and types of chronic infections, and treatment were collected. Patients were treated with either mepolizumab 100 mg every 4 weeks or benralizumab 30 mg every 4 weeks for the first three doses and then every 8 weeks. Both drugs were administered subcutaneously. As all subjects had refractory disease despite optimized maintenance therapy, both drugs were used as part of a targeted and individualized treatment regimen, with no ethical consultation required.

## 3. Results

### 3.1. Observational, Cross-Sectional Study

A total of 249 patients (77.5% female; median (IQR) age: 63.0 (50.0, 71.0) years) were recruited. The median (IQR) levels of blood eosinophils and oral FeNO were 120.0 (80.0, 200.0) cells·µL^−1^ and 14.6 (9.2, 21.9) ppb, respectively. Eighty-eight (35.3%) patients had either eosinophils ≥300 cells·µL^−1^ (53.4%) or FeNO ≥ 25ppb (68.2%) (Figure 1). 19 (9.4%) patients had both eosinophils ≥300 cells·µL^−1^ and FeNO ≥ 25 ppb.

Full description of the total cohort and clinical characteristics according to the two study groups are reported in Appendix A.

Disease severity was significantly higher in bronchiectasis patients with the T2-high endotype in comparison to the rest of the population in terms of median (IQR) level of BSI [6.0 (4.0, 11.5) vs. 5.0 (3.0, 8.0), *p*-value: 0.042]. Bronchiectasis patients with the T2-high endotype had significantly lower values of FEV1%predict, worse dyspnea, and worse quality of life (according to the QoL-B respiration module) compared to the rest of the population, see Table 1).

Patients with both a T2-high endotype and ≥3 annual exacerbations (with both medical and respiratory physiotherapy optimized) were 11 (5.4%).

### 3.2. Severe Asthmatic Patients with Concomitant Bronchiectasis Treated with Anti-IL5 and Anti-IL5-ra

Five bronchiectasis patients with a concomitant diagnosis of severe eosinophilic asthma were treated with anti-IL5 and anti-IL5-ra (i.e., two with benralizumab and three with mepolizumab). Demographics and clinical characteristics of these patients are reported in (Table 2).

Three patients were male, and the median (IQR) age was 63.0 (48.0, 71.0) years. Four patients had idiopathic etiology of bronchiectasis, whereas one had ABPA. Median (IQR) level of peripheral eosinophil counts before treatment was 690.0 (460.0, 700.0) cells·µL^−1^. Median (IQR) length of treatment was 27.0 (24.0, 27.0) months. Exacerbation rate, lung function data, disease severity and use of oral corticosteroids before and after one year and two years of treatment are shown in Table 3 and Figure 2). Exacerbation rates decreased during the year of treatment in comparison to the year before, as well as the prescription of oral corticosteroids. A decreased disease severity was observed during the two-year therapy.

## 4. Discussion

The present research showed that (1) up to one-third of bronchiectasis patients without asthma show a T2-high endotype; (2) bronchiectasis patients with a T2-high endotype have severe disease, mainly related to dyspnea severity, low respiratory function, and high impact of respiratory symptoms on quality of life; (3) 5% of non-asthmatic bronchiectasis patients with a T2-high endotype are frequent exacerbators although both medical and respiratory physiotherapy are optimized, and might be ideal candidates for anti-IL5 and anti-IL5-ra treatments; (4) Treatment with either mepolizumab or benralizumab reduces the exacerbation rate in patients with severe eosinophilic asthma and concomitant bronchiectasis for up to 2 years.

Different definitions of “eosinophilic” or “T2-high endotype” have been provided and very few studies have been conducted in bronchiectasis patients. Sputum eosinophil counts have been used to define airway eosinophilic inflammation across different respiratory diseases with two different cut-offs (3% and 4%) [4,7,20]. Other studies based on blood eosinophil counts or oral FeNO, which can be routinely evaluated, have defined eosinophilic inflammation: They correlate with sputum eosinophilia in several chronic respiratory diseases [4,7,20]. Several cut-offs of eosinophilic blood count have been used, including 3% and 4% or 150 and 300 cells·µL^−1^, also in evaluating patients’ response to ICS. Similar thresholds have been used in asthma and COPD [4,20]. Data of oral FeNO are frequently reported in both COPD and asthmatic patients using two different thresholds, 25 and 50 dpp [4,20]. A clinical approach is used to define the T2-high endotype in bronchiectasis according to criteria used to prescribe biological treatments to patients with severe asthma, which include severe refractory eosinophilic asthma (for mepolizumab), adult patients with severe eosinophilic asthma inadequately controlled despite high-dose inhaled corticosteroids plus long-acting β-agonists with baseline blood eosinophil counts ≥300 cells·µL^−1^ for benralizumab, raised blood eosinophils, and/or FeNO for dupilumab.

Although bronchiectasis is recognized as a neutrophilic inflammatory disease, recent studies reported that one-third of patients show an eosinophilic response [4,20]. Abo-Leya et al. and Viana et al. identified different threshold of peripheral eosinophilia and demonstrated that up to 30% of bronchiectasis patients have >300 blood eosinophils·µL^−1^ [21,22]. Similar thresholds for both eosinophilic and oral FeNO have been used in asthma and COPD to define a patients’ phenotype with high disease severity and frequent exacerbations [4,20].

Two patients (one per study group) with allergic bronchopulmonary aspergillosis (ABPA) were included in the study. All patients underwent ABPA screening including total IgE, blood eosinophils, and IgG and IgE for A. fumigatus. According to previous literature, the incidence of ABPA in Italy is lower than the prevalence in North Europe perhaps because of the latitude factor [23].

We also demonstrated that bronchiectasis patients without asthma and with a T2-high endotype show a more severe disease, in terms of dyspnea, respiratory function, and quality of life. Moreover, we observed an increasing trend of exacerbations in patients with T2 immunity, although these data did not reach statistical significance. Heterogeneous results on the association between the risk of severe exacerbations and blood eosinophilia have been described [21,22]. In order to draw conclusions on the association between exacerbation risk and T2-high endotype in bronchiectasis patients, further large, multicenter studies are needed. 

Recent data underlined that T2 immunity involves both Th2 response and an epithelial–innate lymphoid cell type 2 (ILC2) pathway. Notably, different experiences suggested a possible association between ILC2, and both asthma and COPD, while no data on bronchiectasis have been published so far [24]. Moreover, studies on chronic sinusitis suggested the presence of specific clusters of patients showing an IL-5 driven phenotype with a peculiar expression of other cytokines (e.g., TNFα) underlining the need of better characterizing these patients from a molecular point of view [25].

Chronic *P. aeruginosa* infection has been associated with a severe disease and poor clinical outcomes: We did not find any differences in chronic infection, and specifically in chronic infection by *P. aeruginosa* thresholds, between T2-high and non-T2 high patients [26,27,28,29]. A possible explanation may be the association between *P. aeruginosa* and neutrophilic inflammation [26,27,28,29]. However, eosinophilic inflammation cannot exclude a neutrophilic inflammatory pattern, and this might explain the similarity in chronic infection and exacerbation rate in the two groups. This is supported by previous literature showing that almost half of bronchiectasis patients with sputum eosinophils >3% showed concomitant neutrophilic inflammation in sputum samples [7].

Our case series proved the efficacy of anti-IL5 and anti-IL5ra in terms of reduction of the exacerbations and decreased prescription of oral corticosteroids [30,31]. Other studies described a decreased exacerbation rate in patients treated with either benralizumab or mepolizumab in other chronic respiratory diseases (e.g., asthma and COPD) [32,33].

Unfortunately, we did not evaluate eosinophils in sputum or in other respiratory samples, and we cannot speculate about a correlation between blood and sputum eosinophilia. However, blood eosinophilia is an accepted surrogate of airway eosinophilia in several chronic respiratory diseases, and is included in labels for the prescription of biological drugs [34,35]. Furthermore, we integrated data of blood eosinophils with those from oral FeNO. Another study limitation is the missing collection of data on blood eosinophils and oral FeNO during follow-up. Finally, the monocentric nature of the project can hinder the generalizability of the findings, highlighting the necessity of international data.

One of the study strengths is the high sample size of bronchiectasis patients with blood eosinophil counts and oral FeNO to evaluate a T2-high endotype. Moreover, although we did not assess sputum eosinophils, we adopted a practical and worldwide-accepted definition of T2-high used for the administration of biological drugs. Furthermore, our results suggest the effective role of anti-IL5 and anti-IL5-ra in bronchiectasis patients, with a 2-year follow-up.

Our study findings support a personalized approach in bronchiectasis patients based on specific treatable traits [36]. The T2-high endotype is a treatable trait in different chronic respiratory diseases. Moreover, the identification of an eosinophilic inflammation is key for patients’ stratification and for the selection of specific molecules aimed at treating T2-high endotype.

In conclusion, up to one-third of bronchiectasis adults without a concomitant diagnosis of asthma show a T2-high endotype and are characterized by a more severe disease. A decreased exacerbation rate in bronchiectasis patients with severe eosinophilic asthma treated with benralizumab and mepolizumab was also found. Our findings may be very important in the definition and characterization of a T2-high endotype that may represent a treatable trait in bronchiectasis going toward a personalized medicine approach in bronchiectasis.

## Figures and Tables

**Figure 1 biomedicines-09-00772-f001:**
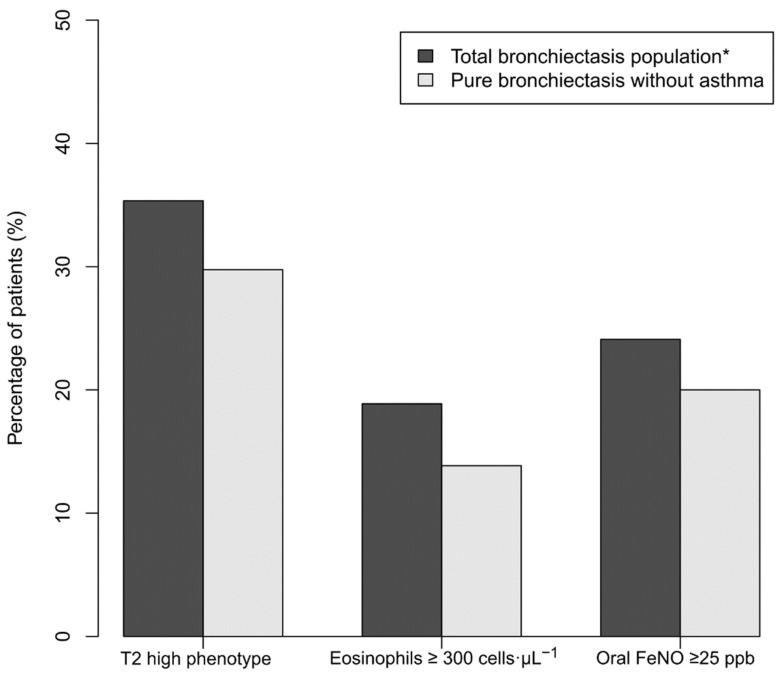
Prevalence of bronchiectasis patients with either T2 high endotype (either eosinophils ≥300 cells·µL^−1^ or oral FeNO ≥ 25 ppb) or eosinophils ≥300 cells·µL^−1^ or oral FeNO ≥ 25 ppb among the entire study population and those without asthma. * Including those with concomitant asthma.

**Figure 2 biomedicines-09-00772-f002:**
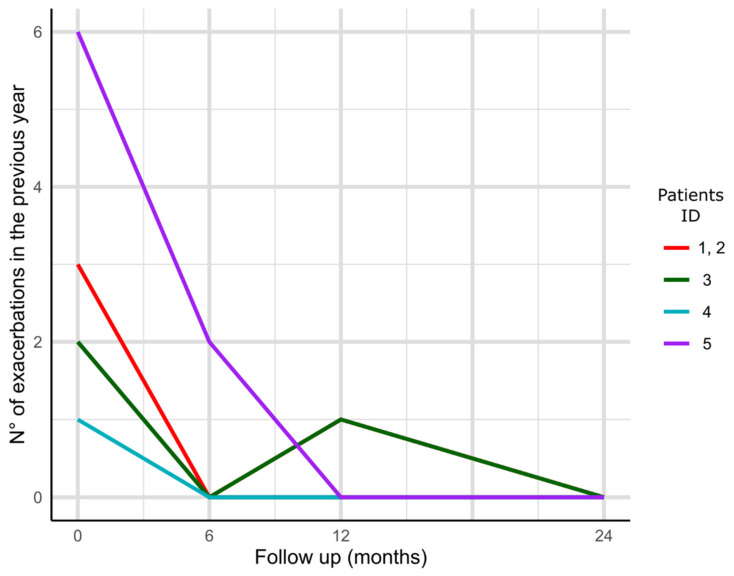
Exacerbation rate of the 5 individual patients up to 24-month follow up. Patient #1 and #2 has the same number of exacerbations.

**Table 1 biomedicines-09-00772-t001:** Clinical characteristics of patients with and without a T2-high phenotype.

Variable		T2-High Endotype (*n* = 63)	Non-T2-High (*n* = 140)	*p*-Value
**Demographics**				
Sex (female), *n* (%)		45 (71.4%)	115 (82.1%)	0.084
Age, median (IQR) years		65.0 (54.0, 70.0)	62.0 (49.5, 71.0)	0.344
Former/Active smokers, *n* (%)		34 (54.0%)	57 (40.7%)	0.079
Body mass index [Kg/m^2^], median (IQR)		21.0 (18.8, 23.6)	21.2 (19.3, 24.0)	0.476
**Radiology**				
Reiff score, median (IQR)		4.5 (3.0, 6.0)	4.0 (2.0, 6.0)	0.097
**Disease severity**				
BSI, median (IQR)		6.0 (4.0, 11.5)	5.0 (3.0, 8.0)	0.042
**Etiology**				
Idiopathic, *n* (%)		39 (61.9%)	86 (61.4%)	0.385
Primary immunodeficiency, *n* (%)		5 (7.9%)	14 (10.0%)	
Post Infective, *n* (%)		4 (6.3%)	15 (10.7%)	
Primary Ciliary Dyskinesia, *n* (%)		2 (3.2%)	7 (5.0%)	
Secondary Immunodeficiency, *n* (%)		4 (6.3%)	5 (3.6%)	
ABPA, *n* (%)		1 (0.7%)	1 (1.6%)	
Other *, *n* (%)		8 (13.6%)	11 (7%)	
**Comorbidities**				
BACI, median (IQR)		0.0 (0.0, 1.0)	0.0 (0.0, 0.0)	0.101
Gastroesophageal reflux disease, *n* (%)		34 (54.0%)	68 (48.6%)	0.477
Cardiovascular diseases, *n* (%)		27 (42.9%)	48 (34.3%)	0.242
Rinosinusithis, *n* (%)		25 (39.7%)	40 (28.6%)	0.116
Osteoporosis, *n* (%)		13 (20.6%)	25 (17.9%)	0.639
Neoplastic disease, *n* (%)		10 (15.9%)	21 (15.0%)	0.873
Depression, *n* (%)		8 (12.7%)	11 (7.9%)	0.273
**Vaccination status**				
Pneumococcal polysaccharide vaccine -23		27 (42.9%)	61 (44.5%)	0.825
Pneumococcal conjugate vaccine -13		43 (68.3%)	95 (69.3%)	0.877
Influenza vaccination during the past year		46 (73.0%)	105 (75.0%)	0.764
**Clinical status**				
Exacerbations in the previous year, median (IQR)		1.5 (0.0, 2.0)	1.0 (0.0, 2.0)	0.099
3+ exacerbations in the previous year, *n* (%)		11 (18.3%)	16 (12.5%)	0.288
>1 hospitalization in the previous year, *n* (%)		8 (13.3%)	9 (7.0%)	0.160
mMRC, *n* (%)	0	30 (48.4%)	92 (65.7%)	0.038
	1	20 (32.3%)	39 (27.9%)	
	2	5 (8.1%)	5 (3.6%)	
	3	4 (6.5%)	3 (2.1%)	
	4	3 (4.8%)	1 (0.7%)	
	3–4	7 (11.3%)	4 (2.9%)	0.015
**Quality of life—QoL-B questionnaire**				
QoL-B questionnaire Physical, median (IQR)		60.0 (40.0, 80.0)	66.7 (40.9, 86.7)	0.330
QoL-B questionnaire Role, median (IQR)		66.7 (46.7, 86.7)	73.3 (53.3, 86.7)	0.321
QoL-B questionnaire Vitality, median (IQR)		55.6 (52.8, 77.8)	55.6 (44.4, 66.7)	0.291
QoL-B questionnaire Emotion, median (IQR)		83.3 (58.3, 100.0)	75.0 (58.3, 91.7)	0.821
QoL-B questionnaire Social, median (IQR)		58.3 (50.0, 83.3)	75.0 (50.0, 91.7)	0.300
QoL-B questionnaire Treatment Burden, median (IQR)		66.7 (66.7, 77.8)	66.7 (54.2, 77.8)	0.395
QoL-B questionnaire Health, median (IQR)		43.0 (22.9, 56.2)	41.7 (25.0, 62.5)	0.468
QoL-B questionnaire Respiration, median (IQR)		66.7 (58.3, 80.6)	77.8 (68.6, 85.2)	0.045
**Functional evaluation**				
FEV_1_, median (IQR) %predict.		80.5 (58.8, 96.0)	86.0 (74.0, 102.0)	0.048
FEV_1_ <50%predict., *n* (%)		8 (13.8%)	5 (3.6%)	0.009
FEV_1_ <35%predict., *n* (%)		4 (7.0%)	2 (1.4%)	0.041
**Microbiology**				
Chronic infection, *n* (%)		22 (40.7%)	52 (44.4%)	0.650
Chronic infection *p. aeruginosa*, *n* (%)		15 (27.8%)	26 (22.2%)	0.429
Chronic Infection *H. Influenzae*, *n* (%)		3 (5.6%)	12 (10.3%)	0.305
Chronic Infection MSSA, *n* (%)		2 (3.7%)	12 (10.3%)	0.146
Chronic Infection MRSA, *n* (%)		1 (1.9%)	3 (2.6%)	0.775
Chronic Infection *S. maltophilia*, *n* (%)		2 (3.7%)	1 (0.9%)	0.187
Chronic Infection *Achromobacter*, *n* (%)		1 (1.9%)	1 (0.9%)	0.573
Chronic Infection *S. pneumoniae*, *n* (%)		0 (0.0%)	1 (0.9%)	0.496
Chronic Infection *A. fumigatus*, *n* (%)		1 (1.9%)	0 (0.0%)	0.140
Other chronic infection, *n* (%)		1 (1.9%)	6 (5.1%)	0.315

Definition of abbreviations: BSI = Bronchiectasis severity index; MRSA = Methicillin-Resistant *Staphylococcus aureus*; MSSA = Methicillin- sensitive *Staphylococcus aureus*; ABPA = Allergic bronchopulmonary aspergillosis. Data are presented as median (interquartile range) or *n* (%). * Connective Tissue Disease, Alpha1-antitrypsin deficiency, Rheumatoid arthritis, Ulcerative colitis, Aspiration, COPD, Inflammatory bowel disease, NTM infection.

**Table 2 biomedicines-09-00772-t002:** Clinical characteristics of patients enrolled at baseline.

Patient	1	2	3	4	5
Biological Drug treatment	Mepolizumab	Mepolizumab	Mepolizumab	Benralizumab	Benralizumab
Treatment duration (months)	41	27	21	27	24
Sex	Male	Male	Female	Female	Male
Age, years	71	78	48	42	63
Aetiology	Idiopathic	ABPA	Idiopathic	Idiopathic	Idiopathic
Asthma-comorbidity	Yes	Yes	Yes	Yes	Yes
Smoking status	Never	Never	Never	Former	Never
BSI	5	7	8	4	12
Exacerbations in the previous year	2	3	2	1	6
Hospitalization in the previous year	0	0	2	0	1
FEV_1_ %predict.	88	80	90	97	72
Blood eosinophils (cells·µL^−1^)	1030	460	690	700	410

Definition of abbreviations: BSI = Bronchiectasis severity index; ABPA = Allergic bronchopulmonary aspergillosis.

**Table 3 biomedicines-09-00772-t003:** Comparison of clinical characteristics of patients at baseline and after 12 months of treatment.

Variable	Baseline	12 Months Follow Up	24 Months Follow Up	*p* Value	*p* Value Baseline vs. 12 Months	*p* Value Baseline vs. 24 Months	*p* Value 12 Mesi vs. 24 Mesi
**Disease severity**							
BSI, median (IQR)	7.0 (5.0, 8.0)	5.0 (4.0, 5.0)	5.0 (4.0, 5.0)	0.232	0.102	0.144	1.000
**Clinical status**							
Exacerbations previous year, median (IQR)	2.0 (2.0, 3.0)	0.0 (0.0, 1.0)	0.0 (0.0, 0.0)	0.007	0.042	0.042	1.000
2+ exacerbations previous year, *n* (%)	4 (80.0%)	0 (0.0%)	0 (0.0%)	0.004	0.048	0.048	1.000
1+ hospitalization previous year, *n* (%)	2 (40.0%)	0 (0.0%)	0 (0.0%)	0.099	0.444	0.444	1.000
Sputum volume (mL), median (IQR)	25.0 (22.5, 27.5)	10.0 (0.0, 10.0)	10.0 (0.0, 20.0)	0.223	0.157	0.317	0.317
Eosinophils blood count cells·µL^−1^, median (IQR)	690.0 (460.0, 700.0)	0.0 (0.0, 0.0)	NA	NA	0.180	NA	NA
**Respiratory function**							
FEV1 % predict., median (IQR)	88.0 (80.0, 90.0)	84.0 (79.8, 91.0)	92.0 (91.0, 100.5)	0.148	0.715	0.109	0.18
FEV1 < 80%, *n* (%)	2 (40.0%)	1 (25.0%)	0 (0.0%)	0.449	1.000	0.464	1.000
**Treatment**							
ICS, *n* (%)	5 (100.0%)	5 (100.0%)	5 (100.0%)	1.000	1.000	1.000	1.000
LABA, *n* (%)	5 (100.0%)	5 (100.0%)	5 (100.0%)	1.000	1.000	1.000	1.000
LAMA, *n* (%)	5 (100.0%)	5 (100.0%)	5 (100.0%)	1.000	1.000	1.000	1.000
OCS, *n* (%)	5 (100.0%)	1 (20.0%)	1 (20.0%)	0.014	0.048	0.048	1.000

Definition of abbreviations: BSI = Bronchiectasis severity index; ICS = inhaled corticosteroids, LABA = Long-Acting Beta-Agonists, LAMA = long-acting muscarinic antagonists, OCS = oral corticosteroids; ABPA = Allergic bronchopulmonary aspergillosis. Data are presented as median (interquartile range) or *n* (%).

## Data Availability

Data are available on request due to privacy restrictions.

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
