# Peer review of "T2-High Endotype and Response to Biological Treatments in Patients with Bronchiectasis"

_biomedicines, 2021, doi:10.3390/biomedicines9070772_

Round 1

Reviewer 1 Report

I thank the authors for sharing their high-quality work . I really enjoyed the paper. The only comment the patient inclusion and exclusion criteria may be defined in more detail. The vaccination status of the patients and other potentials standardizations.

Congratulations 

Author Response

Dear Prof. Pelaia,

The authors would like to thank the Editor for giving them the opportunity to submit a revised version of the paper. We agree with all the comments and recommendations suggested by the reviewers. We have changed the manuscript to comply with reviewers’ recommendations. The following is a detailed response to each of reviewers' recommendations.

Reviewer 1

I thank the authors for sharing their high-quality work. I really enjoyed the paper. The only comment the patient inclusion and exclusion criteria may be defined in more detail. The vaccination status of the patients and other potentials standardizations.

Congratulations

Answer to reviewer 1

We thank the reviewer for the kind comment, and we are glad he/she enjoyed our manuscript.

Concerning the definition of inclusion-exclusion criteria, these were the criteria we used. Nothing more. We decided to enrol the broadest population of bronchiectasis patients seen in our Program to reach the broadest representativity and, thus, generalizability, possible. Concerning the vaccination status, we decided to add this information in Table 1

Reviewer 2 Report

The authors reported the role of type 2 inflammation on bronchiectasis and the effect of biological treatment for the pathophysiology. This study is interesting, however, there are several major concerns with this work as outlined below.

(Major comments)

  1. Recently、it is reported that FeNO and eosinophilic inflammation is associated with not only T helper cell type 2 (Th2) cells but also group 2 innate lymphoid cells (ILC2)( Brusselle GG. et al.: Ann Am Thorac Soc. 11: S322-328, 2014). These are called type 2 inflammation. Therefore, the author should reconsider whether or not to change “Th2” to “Type 2” in the manuscript.
  2. The author should discuss the relationship between ILC2 and this result as mentioned above.
  3. It is reported that there are intimate relations between bronchiectasis and sinusitis, especially type 2 inflammation in the asthmatic airway and eosinophilic sinusitis (Tomassen, P., et al.. "J Allergy Clin Immunol 137(5): 1449-1456 e1444) ,2016). Therefore, the author should discuss this point.

Author Response

Dear Prof. Pelaia,

The authors would like to thank the Editor for giving them the opportunity to submit a revised version of the paper. We agree with all the comments and recommendations suggested by the reviewers. We have changed the manuscript to comply with reviewers’ recommendations. The following is a detailed response to each of reviewers' recommendations.

Reviewer 2

Comment #1

Recently it is reported that FeNO and eosinophilic inflammation is associated with not only T helper cell type 2 (Th2) cells but also group 2 innate lymphoid cells (ILC2) (Brusselle GG. et al.: Ann Am Thorac Soc. 11: S322-328, 2014). These are called type 2 inflammation. Therefore, the author should reconsider whether or not to change “Th2” to “Type 2” in the manuscript.

Answer to comment #1

We thank the reviewer for this comment and we clearly understand that the definition of just Th2 was not broad enough, thus we modified Th2 in Type 2 (T2) throughout the manuscript.

Comment #2

The author should discuss the relationship between ILC2 and this result as mentioned above.

Answer to comment #2

We thank the reviewer for his/her comment and we agree with the need to discuss the relationship between ILC2 and our data.  For this reason, we decided to add the following sentences at line 235-238

Recent data underlined that T2 immunity involves both Th2 response and an epithelial–innate lymphoid cell type 2 (ILC2) pathway. Notably, different experiences suggested a possible association between ILC2, and both asthma and COPD, while no data on bronchiectasis have been published so far. [24]

Comment #3

It is reported that there are intimate relations between bronchiectasis and sinusitis, especially type 2 inflammation in the asthmatic airway and eosinophilic sinusitis (Tomassen, P., et al.. "J Allergy Clin Immunol 137(5): 1449-1456 e1444) ,2016). Therefore, the author should discuss this point.

Answer to comment #3

We thank the reviewer for his/her comment and we understand the need to further discuss about t2 inflammation. For this reason, we decided to add the following sentence at line 239-242:

Moreover, studies on chronic sinusitis suggested the presence of specific clusters of patients showing an IL-5 driven phenotype with a peculiar expression of other cytokines (e.g. TNFα) underlining the need of better characterizing these patients from a molecular point of view [25].

Round 2

Reviewer 2 Report

Revision manuscript is well written. This study would provide a new insight for difficult type 2 airway inflammation.

Author Response

The authors would like to thank the editor and the reviewers for the suggestions that significantly improved the manuscript and to have the manuscript published on Biomedicines